# The 50th Anniversary Conference − Caxambu 2024

**Renata Rosito Tonelli**[1]/+, **Daniel José Galafasse Lahr**[2], **Elvira Maria Saraiva**[3],
**Angela Hampshire de Carvalho Santos Lopes**[3]

[1]Universidade Federal de São Paulo, Instituto de Ciências Ambientais, Químicas e Farmacêuticas,
 Departamento de Ciências Farmacêuticas, Diadema, SP, Brasil
[2]Universidade de São Paulo, Instituto de Biociências, Departamento de Zoologia, São Paulo, SP, Brasil
[3]Universidade Federal do Rio de Janeiro, Instituto de Microbiologia Paulo de Góes, Rio de Janeiro, RJ, Brasil

Chagas disease (CD), caused by *Trypanosoma* (*Schizotrypanum*) *cruzi*, remains a major global health concern, particularly in Latin America, where millions are at risk. To mark five decades of Chagas research, the Brazilian Society of Protozoology (SBPz) hosted a four-day conference held in Caxambu, Minas Gerais, Brazil, from November 3 to 7, 2024. The meeting brought together world-renowned experts from diverse disciplines whose work has significantly advanced the boundaries of CD studies. Key discussions focused on the parasite's genetic and metabolic adaptability, with special emphasis on genomic compartmentalisation, RNA processing, and metabolic flexibility essential for survival and pathogenesis. New insights into host-parasite interactions highlighted inflammatory and vascular remodelling processes that drive parasite dissemination and disease progression, especially in cardiac tissue. In the area of drug development, researchers noted treatment limitations, the urgency for novel therapeutic candidates, and ongoing clinical trials assessing alternative regimens of benznidazole (BZN) and nifurtimox (NFX). Progress in biomarker discovery and vaccine development was also discussed as pivotal to improving disease diagnosis, prognosis, and prevention. Beyond laboratory research, the meeting highlighted the importance of science communication and public health engagement. Outreach initiatives and educational exhibitions were showcased as tools to raise awareness and enhance access to disease diagnosis and treatment. Altogether the integration of multidisciplinary approaches from molecular biology to public policy underscores the enduring commitment to combating CD through research, collaboration, and innovation.

Key words: Chagas disease - *Trypanosoma cruzi* - pathogenesis - drug development - public health strategies

Chagas disease (CD), or American trypanosomiasis is a potentially life-threatening parasitic infection caused by the protozoan *Trypanosoma* (*Schizotrypanum*) *cruzi*. While the disease is endemic in 21 Latin American countries, its global relevance has expanded in recent decades due to increased human mobility and migration. According to the World Health Organization (WHO), an estimated 6 to 7 million individuals are currently infected and approximately 75 million people worldwide remain at risk of acquiring the infection.[1]

The primary mode of transmission is vectorial, through contact with the faeces and urine of triatomine bugs harbouring *T. cruzi*. However, oral transmission via ingestion of contaminated food and beverages has become increasingly recognised as a significant epidemiological pathway, particularly in regions where vector control has reduced domiciliary transmission, posing new challenges for disease control. CD manifests in two clinical stages.[2] The acute phase, which typically lasts between four to eight weeks, is often mild or asymptomatic, thereby complicating early diagnosis. In the absence of treatment, the infection can persist into a chronic phase, during which approximately 30% to 40% of the individuals develop severe cardiomyopathy, gastrointestinal megasyndromes or other systemic complications requiring specialised treatment. Whilst treatment may lower the risk of disease progression, its effectiveness remains inconsistent and not fully understood. Despite the disease's substantial burden in terms of morbidity, mortality and the socioeconomic impact, diagnostic and treatment coverage remain critically low: fewer than 10% of affected individuals are diagnosed, and only about 1% receive antiparasitic treatment.[3]

## Plenary lecture (opening conference)

*Fifteen years of a successful proposal: T. cruzi strains should be assigned to one of the discrete typing units* - In this plenary presentation, Bianca Zingales (Universidade de São Paulo, São Paulo, Brazil) provided a comprehensive overview of the genetic and phenotypic diversity of T. cruzi, the causative agent of CD.[4] Her talk underscored the profound complexity of this protozoan parasite and its relevance to disease outcomes, diagnosis, and treatment strategies.

Financial support: CAPES, CNPq, FAPESP, FAPERJ, INCTEM, The Company of Biologists, and the International Society of Protistologists - ISOP.
+ Corresponding author: r.tonelli@unifesp.br
 https://orcid.org/0000-0001-8495-2073

Dr Zingales highlighted the historical progression of T. cruzi classification systems, culminating in the current consensus framework established in 2009, which recognises six discrete typing units (DTUs), TcI to TcVI, with the subsequent addition of TcBat. She emphasised the significance of genomic studies that have revealed TcV and TcVI as hybrid lineages and documented natural genetic exchange within the species. The presentation further explored how antigenic diversity, particularly in surface proteins, facilitates immune evasion and poses a major obstacle for vaccine development. She discussed the geographical distribution of DTUs, noting that specific lineages tend to predominate in certain regions (*e.g.*, TcI in Central America and the USA; TcII, TcV, and TcVI in Bolivia). Importantly, associations between DTUs and clinical manifestations were addressed, including the prevalence of cardiac complications with TcI and digestive forms with TcII.

She concluded by addressing the challenges posed by this diversity to treatment efforts. Although benznidazole (BZN) and nifurtimox (NFX) remain the standard therapeutic agents, their efficacy appears independent of DTU classification. The speaker called for the establishment of representative, well-characterised strain panels to inform the development of novel diagnostics, vaccines, and therapeutics capable of addressing the broad biological variability of *T. cruzi*.

### Mechanisms of *T. cruzi* pathogenesis

*Trypanosoma cruzi* is a stealth pathogen that manipulates host tissues to facilitate its survival and dissemination. During infection, the parasite migrates through inflamed tissues, in search of permissive host cells, either macrophages or non-phagocytic cells, where it undergoes intracellular replication.[5] This process is not silent; intravital microscopy has revealed that microvascular leakage occurs early in infection, triggered by inflammatory signals that open endothelial "floodgates". This leakage delivers essential nutrients to infected cells, sustaining parasite metabolism, inhibits apoptosis, and contributes to systemic dissemination as host cells rupture and release trypomastigotes. The parasite exploits newly formed microvessels[6] by engaging endothelial-binding motifs from trans-sialidase (TS) antigens. Research led by Julio Scharfstein's group (Universidade Federal do Rio de Janeiro - Rio de Janeiro, Brazil) has identified angiogenesis as a hallmark of *T. cruzi* infection, mediated by chymase, a serine protease from mast cells.[6] This mechanism resembles non-canonical tumor-induced angiogenesis observed in endothelial TLR2-deficient mice, where oxidative lipids promote vascular remodelling. In parallel, extracellular vesicles continuously released by trypomastigotes[7] are enriched with tGPI-mucins, potent TLR2 activators that may influence inflammatory neovascularisation. The interaction between the mast cell/chymase pathway and the kallikrein-kinin system (KKS) suggests a broader inflammatory mechanism that bridges innate and adaptive immunity in CD.[8]

*Trypanosoma cruzi* infection elicits a strong inflammatory response, characterised by the elevated production of reactive oxygen species (ROS) and nitric oxide (NO). While essential for parasitic control, excessive ROS and NO can cause collateral tissue damage. This is particularly evident in the heart, where *T. cruzi* infection by leads to severe cardiac pathology, including arrhythmias, heart failure, and microvascular abnormalities.[3] Artur Santos Miranda (Universidade Federal de Minas Gerais, Minas Gerais, Brazil) addressed the roles of oxidative stress and disrupted calcium signalling as key factors influencing *T. cruzi* infectivity and the development of host cardiac Chagasic cardiomyopathy (CCM). His research highlights the enzyme CaMKII, a key regulator of calcium homeostasis and oxidative balance, as a potential therapeutic target. *In vitro* and *ex vivo* studies demonstrate that inhibition of the $Ca^{2+}$/CaM-CaMKII pathway reverses arrhythmic profiles of isolated hearts and isolated left-ventricles cardiomyocytes.[9] Miranda proposes that dual inhibition CaMKII in both parasite and host could impair parasite replication, reduce inflammation, and prevent cardiac remodelling, ultimately improving disease outcomes.

This multidisciplinary provides new insights into the cellular and molecular mechanisms underpinning *T. cruzi* infection and CCM progression, and points to novel therapeutic strategies that simultaneously target both the parasite and host dysfunction.

### Tackling CD

CD remains a critical global health issue. Despite significant progress in recent years, major challenges persist, particularly in drug development and treatment efficacy. A range of complementary approaches now offers a comprehensive view of current advances and future directions in the fight against this neglected tropical disease.[10]

BZN and NFX, the only two drugs approved by the Brazilian Ministry of Health for CD therapy, have been in use for over half a century.[11,12] While these are effective during the acute phase and some chronic cases, they present serious limitations. Treatment regimens are lengthy (60-90 days) and are frequently associated with severe side effects, leading to high rates of discontinuation. Moreover, their efficacy in the chronic phase, where most patients are diagnosed, is limited.[13] Jadel Müller Kratz (Drugs for Neglected Diseases Initiative, Rio de Janeiro, Brazil) discussed current challenges in developing new, safer and more accessible treatments, for both the acute and chronic stages. His presentation also addressed the translational hurdles involved in drug development, the need to expand access to diagnosis and treatment, the importance of identifying improved biomarkers to assess treatment response and predict disease progression, and the pressing need for greater coordination and investment in the field.[14]

Recent advances in technology have driven progress in drug discovery. Carolina Borsoi Moraes (Universidade de São Paulo, São Paulo, Brazil) highlighted the role, of automated phenotypic screening and sensitive *in vivo* models.[15] Phenotypic screening remains a cor-

nerstone of antiparasitic drug discovery,[16] enabling the rapid testing of large compound libraries for antiparasitic activity. This has not only accelerated candidate identification but also provided valuable insights into mechanisms of action. Nevertheless, significant barriers remain, such as the parasite's ability to persist despite treatment and the continued absence of reliable, clinically validated biomarkers for cure or disease progression.

In this context, Igor Correia de Almeida (University of Texas at El Paso, Texas, USA) presented the TESEO clinical trial (New ThErapies and Biomarkers for ChagaS infEctiOn) that represents a major step forward. With six treatment arms and 450 patients, the TESEO trial is evaluating novel dosing regimens of BZN and NFX, with the aim of identifying strategies that are both safer and more effective.[17] The trial incorporates the use of molecular biomarkers such as quantitative polymerase chain reaction (qPCR) to improve assessment of treatment response and disease evolution. Importantly these trials underscore the need of tailoring treatments to regional differences in therapeutic responses and advancing biomarker research to predict long-term outcomes.

Ricardo Toshio Fujiwara's (Universidade Federal de Minas Gerais, Minas Gerais, Brazil) explored the feasibility of a pan-Trypanosomatidae vaccine targeting multiple kinetoplastid parasites, including *T. cruzi*, *Leishmania infantum*, and *Leishmania mexicana*. Capitalising on genetic similarities within the Kinetoplastida class, his team developed a broad-spectrum chimeric polyprotein vaccine incorporating CD8+ T cell epitopes conserved across *Leishmania* spp. and *T. cruzi*.[18] Immunisation with this broad-spectrum vaccine induced specific IgG production, reduced parasite load and decreased inflammation in the colon, fewer degenerated hepatocytes, and increased connective tissue proliferation in the skin lesions. These findings support the concept of a pan-Trypanosomatidae vaccine, paving the way for a new generation capable of targeting diverse infectious agents within this parasite family.

Together, these studies underscore the multifaceted approach required to combat CD, encompassing basic research, drug discovery, translational medicine, clinical trials, and vaccine development. Continued investment and interdisciplinary collaboration are essential to overcome treatment limitations, improve disease management, and advance toward effective control strategies.

## Metabolic adaptability of *T. cruzi*

Several presentations converged on a central theme: the remarkable metabolic adaptability of *T. cruzi*. Each research effort explored distinct aspects of the parasite's energy metabolism, emphasising emphasising its ability to exploit multiple biochemical pathways to survive in the diverse environments encountered throughout its life cycle. Ariel Mariano Silber (Universidade de São Paulo, São Paulo, Brazil) focused on *T. cruzi's* ability to utilise histidine as an energy source. His work demonstrated that the parasite efficiently converts this amino acid into ATP through mitochondrial oxidation, highlighting its metabolic flexibility and ability to shift between carbohydrate and amino acid metabolism in response to environmental cues.[19]

Anibal Vercesi (Universidade de Campinas, Sao Paulo, Brazil) explored the intricate interplay between mitochondria and acidocalcisomes in regulating calcium homeostasis. She underscored the importance of mitochondrial calcium ion ($Ca^{2+}$) uptake in buffering cytosolic $Ca^{2+}$ levels and regulating bioenergetics, oxidative phosphorylation, and redox balance. These mechanisms are critical for controlling autophagy and programmed cell death. Central to this process is the mitochondrial $Ca^{2+}$ uniporter (MCU), whose initial identification in trypanosomes contributed to the elucidation of its molecular structure in all eukaryotes.[20] This research provides new insights into This research provides valuable insights into the composition, function, and physiological role of the MCU in *T. cruzi* and points to new avenues for therapeutic intervention.

*Trypanosoma cruzi* relies on reduced Nicotinamide Adenine Dinucleotide Phosphate (NADPH) as a crucial cofactor for lipid and nucleic acid synthesis, as well as oxidative stress defence. Essential for parasite growth and survival, NADPH is primarily produced through the oxidative branch of the pentose phosphate pathway and enzymes associated with the citric acid cycle.[21] Arthur Torres Cordeiro's (Centro Nacional de Pesquisas em Energia e Materiais de Campinas, São Paulo, Brazil) research took a chemical biology approach, exploring how inhibitors targeting key metabolic enzymes can disrupt *T. cruzi* metabolism. He identified inhibitors of enzymes such as glucose-6-phosphate dehydrogenase and malic enzyme, offering promising targets to disrupt the parasite's metabolic machinery and. viability.

Marcia Cristina Paes (Universidade Estadual do Rio de Janeiro, Rio de Janeiro, Brazil) contributed to understanding the metabolic integration between mitochondria and glycosomes, showing how *T. cruzi* adapts its energy production based on available nutrients. Her findings demonstrated that in glucose-rich environments, *T. cruzi* downregulates mitochondrial activity in favour of succinate fermentation. This metabolic shift enhances survival under variable oxygen conditions, illustrating yet another layer of the parasite's adaptive capacity.[22]

Collectively, these studies illuminate the extraordinary metabolic plasticity of T. *cruzi*, highlighting its capacity to adjust and fine tune iris energy production pathways, exploit host environments, and ensure survival across different stages and hosts. Whether through amino acid utilisation, organelle interactions, enzyme inhibition, or metabolic shifts, each contribution advances our understanding of the parasite's physiology and identifies potential metabolic vulnerabilities.

## Key insights into *T. cruzi* biology

Fundamental aspects of *T. cruzi* biology were explored by six researchers. Maria Carolina Elias (Instituto Butantan, São Paulo, Brazil), discussed the genomic architecture of *T. cruzi* noting its compartmentalised genome. The core regions contain conserved genes, while in disruptive regions are enriched in rapidly evolving multigenic families, potentially enhancing the parasite's infectivity.[23] DNA replication origins were found more frequently in four regions, particularly within DGF-1

genes, whereas Core regions appear to lack Orc1Cdc6, a key component of the prereplication complex. Single nucleotide polymorphism (SNP) analysis suggests that DNA replication contributes to genetic variability, thereby promoting adaptability and survival.

Sergio Schenkman (Universidade Federal de São Paulo, São Paulo, Brazil) presented transcriptomic analyses, revealing high transcriptional variability in *T. cruzi*, especially within multigene families. These findings are consistent with previous RNA-seq studies that showed distinct gene expression profiles between virulent and non-virulent T. cruzi strains, with particular emphasis on surface protein genes involved in host interaction.[24] BZN treatment was shown to alter the transcriptomic profile of intracellular amastigotes, potentially leading to the enrichment of a latent, drug-resistant cell subpopulation. However, issues such as genome variability and host cell RNA contamination highlight the need to optimise of parasite isolation methods to improve transcriptomic study accuracy. Building upon these insights, Irina Afasizheva (University of California, Irvine, USA) explored mitochondrial RNA processing in trypanosomes.[24] Her team identified the mitochondrial 3′ processome (MPsome), a complex consisting of KRET1 TUTase, KDSS1 3′-5′ exonuclease, and six structural proteins involved in RNA maturation and degradation. The work highlights the role of a DEAD/H-box helicase (KREH3) in processing double-stranded RNA and stabilising guide RNAs. Although focused on T. brucei, these mechanisms may offer important insights for *T. cruzi* research.[25]

José Roberto Sotelo-Silveira (Universidad de la República, Montevideo, Uruguay) investigated the translational machinery of *T. cruzi*, showing that ribosomal protein (RP) mRNA translation is globally repressed during the metacyclic stage, though some RP mRNAs remain active, possibly indicating ribosome specialisation. Additionally, ribosome-associated non-coding RNAs (rancRNAs) derived from snoRNAs and snRNAs appear to modulate translation across different life cycle stages,[26] highlighting the dynamic regulation of protein synthesis in response environmental changes.

The parasite's adaptive mechanisms to environmental stresses are illustrated when replicative epimastigotes differentiate into non-proliferative metacyclic trypomastigotes (metacyclogenesis) when nutrients are scarce, and growth is impaired.[27] Simone Guedes Calderano (Instituto Butantan, São Paulo, Brazil) examined cell cycle dynamics during metacyclogenesis. Most stationary epimastigotes were arrested in the G1 phase (~75%). Expression patterns of six key proteins revealed that cyclin-dependent kinases (CRK1 and CRK2) and DNA replication factors (MCM6 and MCM7) were expressed throughout the cell cycle while cyclin five and Wee1 displayed phase-specific expression peaking at the G1/S transition and Wee1 at the S phase. During metacyclogenesis, CRK3, Wee1, MCM6, and MCM7 expression levels declined, while CRK1 and cyclin five remained active, suggesting a tightly regulated transition that halts proliferation and promotes differentiation.[28]

Studies using cryo-electron microscopy (cryo-EM) and cryo-electron tomography (cryo-ET) have provided unprecedented insights into the parasite's osmoregulatory system, critical for parasite's survival. Ingrid Augusto (Universidade Federal do Rio de Janeiro, Rio de Janeiro, Brazil) presented the current knowledge on the crucial role of the contractile vacuole complex (CVC) and acidocalcisome in maintaining *T. cruzi*'s osmotic balance under ionic and osmotic stress conditions.[29] Acidocalcisome is an electron-dense organelle that stores polyphosphates and cations.[30] Cryo-ET revealed membrane proteins in the CVC and confirmed the presence of fusion pores, supporting previous findings from frozen-substituted samples. Additionally, acidocalcisomes displayed varying structural organisations and were observed in contact with mitochondria, suggesting possible ion exchange mechanisms.

Once again *T. cruzi* remarkable adaptability of across its life cycle is shown, emphasising the complex interplay between genetic regulation, cell cycle control, and structural dynamics.

## Bridging science and society: CD awareness initiatives

For the first time in the 50-year history of the Caxambu meeting, four scientific initiatives on CD were integrated. These activities showcased innovative to raise awareness of CD, its history, and ongoing scientific contributions to its study and control.

Luiz Antonio Botelho Andrade (Universidade Federal Fluminense, Rio de Janeiro, Brazil) presented creative methods to engage schoolchildren in learning about the discovery of CD. His approach incorporates playful and interactive educational tools, such as historical models and stop-motion short films that feature key locations linked to Carlos Chagas' pioneering work. Developed by the Scientific Audiovisual Laboratory at Universidade Federal Fluminense, this initiative is part of the Integra Chagas Brazil Project (www.integrachagasBrazil.org), and it has been extended to schools in Caxambu with support from SBPz.

Another notable initiative, Expresso Chagas or EC (https://expressochagas.com/), presented by Tania Cremonini de Araújo-Jorge (Instituto Oswaldo Cruz, Rio de Janeiro, Brazil) was social technology project, developed by the Oswaldo Cruz Institute in collaboration with the Rio Chagas Association. EC uses the concept of an 'imaginary train' guided by Carlos Chagas to blend science and art as a vehicle for education.[31] Since its launch, EC has undertaken expeditions in 11 cities, demonstrating technical, social and political viability. The initiative raises public awareness about transmission, prevention, and treatment options, whilst encouraging community engagement in health promotion. To further its reach, EC offers both in-person and virtual training courses for interdisciplinary teams, enabling them to tailor and implement the program in different local settings. Expresso Chagas is aligned with the Brazilian Ministry of Health's 'Brazil Saudável' initiative (https://www.gov.br/saude/pt-br/composicao/saps/Brazil-saudavel), which addresses diseases driven by social determinants.

Added to this there was an engaging interactive game *Escape Room: the lassance enigma* (https://fiocruz.br/escape-room-da-ciencia-Brazileira-1a-parte-descobertas-de-carlos-chagas) developed by Eduardo Caio Torres dos Santos (Instituto Oswaldo Cruz, Rio de Janeiro, Brazil). Inspired by Carlos Chagas' 1909 expedition to the rural town of Lassance, Minas Gerais, where he identified the unknown disease that led him to make one of the most ground-breaking medical discoveries. The game invites the participants to travel back in time and investigate a medical mystery. The game follows an escape room format, giving players one hour to search for clues, unlock padlocks and solve challenges. Aimed at students and general audiences, the experience fosters curiosity, teamwork and inspires scientific discovery.

Complementing these efforts, the exhibition "A Look at the Past and Present of Chagas Disease" was held in Praça 16 de Setembro, in Caxambu. Conceived by Renata Rosito Tonelli (Universidade Federal de São Paulo, São Paulo, Brazil), Júlia Pinheiro Chagas da Cunha (Instituto Butantan, São Paulo, Brazil), and Maisa Splendore Della Casa (Instituto Butantan, São Paulo, Brazil), the exhibit featured historical photographs and documents from the collection of the Museu de Saúde Pública do Emilio Ribas (https://parquedaciencia.butantan.gov.br/programacao/atracoes/museu-de-saude-publica-emilio-ribas), as well as 3D models of *T. cruzi*, kissing bug specimens, and a microscope to promote hands-on-learning The event was supported by the communication team of the Butantan Institute, funded by "*Loccus do Brazil*" (https://www.loccus.com.br) and assisted by graduate student volunteers. Held one day before the annual meeting, the exhibition attracted over 60 visitors of varying age groups and backgrounds, offering them an accessible introduction to a deeper understanding of CD, its causative agent (*T. cruzi*) and its vector from its discovery to the present day.

Together, these outreach efforts underscore the importance of bridging scientific knowledge and public engagement. By integrating education, creativity and community participation, these initiatives strengthen public health awareness and reinforce the value of science addressing neglected disease like CD.

### A legacy of scientific excellence: the research contributions of Professor Erney Plessmann de Camargo

As part of the celebrations, a symposium was held to honour the life and legacy of Professor Erney Plessmann de Camargo (★1935 †2023), a distinguished scientist whose career significantly shaped parasitology and microbiology in Brazil and beyond. Professor Jeffrey Shaw (Universidade de São Paulo, São Paulo, Brazil) asserted that Professor Erney Camargo was a trailblazer in parasitology and microbiology, pioneering interdisciplinary studies on emerging vector-borne parasitic diseases such as malaria, medical and veterinary trypanosomiasis, leishmaniasis, and bacterial infections like spotted fever.[32,33,34,35] One of his most well-known contributions to the study of CD research was the introduction in 1964 of a liquid medium (LIT) for the cultivation of the parasite. His work underscored the importance of a One Health approach, integrating molecular epidemiology, phylogenetics, and taxonomy to deepen our understanding of parasite biodiversity, evolutionary relationships, transmission dynamics, and host-vector interactions.[32,36,37,38,39] His molecular studies on trypanosomatids were instrumental in elucidating their genetic diversity, classification, and life cycles — contributions with direct implications for disease control in human and veterinary medicine.[36,40,41,42]

A tribute by João Marcelo Pereira Alves (Universidade de São Paulo, São Paulo, Brazil), highlighted Professor Camargo's contributions to the study of Strigomonadinae trypanosomatids.[43] These protozoan parasites are characterised by their long-standing endosymbiotic relationship with betaproteobacteria, a symbiosis that evolved over 90 million years ago. Professor Camargo played a pivotal role in uncovering the genetic and biochemical foundations of this association, showing how the endosymbionts supply heme, essential amino acids, and vitamins crucial for the host parasite's survival.[44] Through the application of microscopy, biochemistry, and genomic sequencing, he and his colleagues demonstrated how the metabolic pathways of host and symbiont are functionally integrated. This ground-breaking research remains foundational to our understanding of host-microbe interactions at the molecular level.[43,44,45]

In her tribute, Professor Lucile Maria Floeter-Winter (Universidade de São Paulo, São Paulo, Brazil) emphasised that Professor Erney Camargo was more than just a scientist — he was an educator, mentor, and scientific leader. His influence extended beyond his research, inspiring generations of students and colleagues at USP's Institute of Biomedical Sciences (ICB). His leadership in national scientific institutions, including the Brazilian Academy of Sciences, the Brazilian Society of Protozoology, the Brazilian Society of Tropical Medicine and the National Council for Scientific and Technological Development (CNPq) demonstrated his commitment to advancing science in Brazil. Beyond academia, he was a staunch advocate for public health policy and the strategic translation of scientific knowledge into effective disease control measures.

Professor Erney Camargo received numerous national and international honours in recognition of his extraordinary contributions to protozoology, parasitology, and molecular biology. His scientific legacy endures, not only through his publications and discoveries, but also in the generation of researchers he mentored and inspired. As Professor Floeter-Winter aptly stated, Professor Erney Camargo was a man ahead of his time, always seeking action and directing his ideas toward advancing science and education.

### Historical significance of the CD meeting

During the celebrations, Samuel Goldenberg (Fundação Oswaldo Cruz, Curitiba, Brazil) and Walter Colli (Universidade de São Paulo, São Paulo, Brazil) reflected on the historical significance and impact of the Caxambu CD meeting, which began in 1974. They approached the subject from complementary perspectives.

Samuel Goldenberg analysed scientific publications, including master's dissertations and doctoral theses related to CD and its causative agent, *T. cruzi*, in parallel Walter Colli offered a detailed account of the meeting's early years, highlighting key milestones from 1974 to 1975. The year 1974 marked the first effort to unite Brazilian protozoologist, with an initial meeting at CNPq in Rio de Janeiro. In 1975 the inaugural annual meeting on basic research in CD, took place at Hotel Glória in Caxambu, Minas Gerais, organised by Zigman Brener — one of Brazil's most eminent parasitologists. The scientific and institutional history of protozoology in Brazil was further explored in an article by Goldenberg et al.[46] which underscore how the 1974 meeting catalysed methodological standardisation, fostered interdisciplinary collaboration, and positioned Brazil as a leader in parasitic disease research.

Over the decades, advances in medicine and biotechnology have reshaped the research agenda in protozoology. While the early years focused on developing and disseminating experimental methods, contemporary research now prioritises molecular investigations of *T. cruzi*, as well as the study of pathophysiological mechanisms, diagnostics, treatment, and prevention of CD (Figure).

*In conclusion* - In the 1970s, CD posed a significant public health threat in Brazil, particularly in rural areas. At that time, national control programmes were still in their infancy. The agenda for CD in Brazil at that period, according to the 2015 Brazilian Consensus on Chagas Disease[47] — developed in collaboration between the Brazilian Society of Tropical Medicine and the Ministry of Health — the national agenda focused on vector control through insecticide spraying (DDT), ensuring blood transfusion safety, expanding of epidemiological surveillance, promoting research, and introducing early treatment strategies with NFX (launched by Bayer in 1965) and BZN (launched by Roche in 1971).[11,12]

Over the past five decades, this Caxambu meeting has evolved into a premier international platform for CD research. It brings together senior experts, early-career researchers, and students to share insights and explore future directions in the fight against CD.

This year's discussions reaffirmed the multidisciplinary nature of Chagas research, encompassing genomics, parasite metabolism, pathogenesis, drug development, and public health interventions. Highlights included major discoveries in the genetic diversity of *T. cruzi*, new molecular mechanisms of parasite survival, and innovative therapeutic strategies. The incorporation of cutting-edge technologies, such as cryo-electron tomography, multi-omics analyses, and clinical trials, is driving transformative progress in diagnosis, treatments, and vaccine development.

Beyond scientific advancement, the meeting also underscored the importance of community engagement and science communication. Initiatives like Expresso Chagas XXI, educational exhibitions, and interactive learning activities exemplify the researchers' commit-

## The Agenda 50 Years Later, 2024

**Chagas Disease Research Agenda, 1974**

1) Characterisation of *T. cruzi* in laboratory conditions

2) In vitro maintenance of *T. cruzi*

3) Growth and differentiation of *T. cruzi* in culture media

4) In vivo maintenace of *T. cruzi* in animal models

5) Nutritional requirements of *T. cruzi*

6) Variations between *T. cruzi* strains

7) Methods for the cryopreseravtion of *T. cruzi*

8) *T. cruzi* strain repository and reference center

**Advances in *T. cruzi* Biology, Adaptation and Environmental Responses**

**Metabolic Adaptability of T. cruzi**
TTo assess the parasite's metabolic flexibility in response to environmental changes and its reliance on specific nutrients for survival.

**Fundamental Aspects of *T. cruzi* Biology**
To explore the genomic structure, transcriptional variation, and RNA processing driving the parasite's adaptability and persistence.

To investigate key cell cycle regulators and epigenetic modifications driving parasite proliferation and differentiation.

To describe the molecular and cellular processes underlying *T. cruzi* differentiation, growth, and survival in diverse hosts.

To analyse how the parasite maintains ionic and osmotic balance to adapt to host-induced stress.

**Chagas Disease Diagnosis, Treatment, and Drug Development**

**Mechanisms of *T. cruzi* Pathogenesis**
To understand parasite–host immune interactions and their impact on disease outcome.

**Drug Discovery and Associated Challenges**
To identify and characterise targets for therapy and vaccine development.

**Clinical Trials and Advances in Biomarkers for Chagas Disease**
To review new diagnostics approaches, ongoing therapeutic trials, and the use of biomarkers to monitor disease progression and treatment responses.

**Vaccine Development for *T. cruzi***
To assess the flexibility of developing a multi-kinetoplastid vaccine targeting T. cruzi and related pathogens.

**Public Engagement, Health Policy and Education in Chagas Disease**

**Scientific Outreach and Awareness Initiatives**
To present initiatives aimed at educating communities and engaging the public through interative learning methods

**Chagas Disease in Public Health and Policy**
To anlyse healthcare accessibility, policy-making, and implementation strategiesto improve global management of Chagas Disease.

Comparison of the Chagas Disease Research Agenda from the 1970s and 2024. The 1974 agenda focused on fundamental aspects of *Trypanosoma cruzi* biology, including strain characterisation, *in vitro* and *in vivo* maintenance, growth, differentiation, and nutrition. In contrast the 2024 Agenda expands to include environmental adaptation, host-pathogen interactions, novel diagnostics, drug development, vaccine research, and public health initiatives. This evolution highlights the progress in understanding *T. cruzi* and addressing Chagas disease as a global health challenge.

ment to bridging the gap between science and society, ensuring that research findings lead to tangible improvements in affected communities.

Looking ahead, CD remains a global health priority. Continued investment, interdisciplinary collaboration, and evidence-based policy action are critical to overcoming persistent barriers in diagnosis, treatment, and prevention. By bringing together scientists, healthcare professionals, and policymakers, the Caxambu meeting continues to serve as a catalyst for innovation, accelerating both scientific discovery and public health impact in the ongoing effort to reduce the burden of CD worldwide.

### ACKNOWLEDGEMENTS

To Ana Paula Lopes Vidal and Vilma Araújo for their dedication and unwavering support in organising the SBPz congresses, as well as Cynthia Sayuri Bando and Raquel Vedovato Veras e Silva for their technical assistance. The authors are grateful to Samuel Goldenberg, Bianca S Zingales and Walter Colli for kindly sharing their unpublished manuscript, "Basic research on Chagas disease: fifty years of a successful initiative", which provided valuable insights for this work. The authors also wish to express their sincere gratitude to Professor Jeffery Shaw for his invaluable contribution to the preparation of this article. His careful attention to the English language was fundamental to the final version.

### AUTHORS' CONTRIBUTION

The manuscript was conceived by RRT, and all authors - RRT, DJGL, EMS and AHCSL - contributed to its writing and critically revised the final version.

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

# OPEN PEER REVIEW

Memórias do IOC thanks the anonymous reviewers for their contribution to the peer review of this work.

## FIRST REVIEW ROUND

**REVIEWERS' COMMENTS**

### REVIEWER #1

This manuscript is a bit long for the "perpsective article" format currently adopted by Memórias do IOC. However, I have some suggestions to adjust the text as a "perspective article".

1) topic "The 50th Anniversary Conference - Caxambu 2024":

a) *I suggest removing the entire topic from the main text of article, and include the topic title into the main article title.*

b) The paragraph "To celebrate 50 years of Chagas research, the Brazilian Society of Protozoology (SBPz) organized a four-day conference held in Caxambu, Minas Gerais, Brazil, from November 3 to 7, 2024. The meeting gathered world-leading experts from multiple disciplines whose research has expanded the boundaries of Chagas Disease studies." *Please, add this paragraph to the abstract.*

2) Benznidazole and Nifurtimox: *Please add references to the original works proposing the use of these drugs in Chagas Disease therapy.*

3) topic "Mechanisms of T. cruzi Pathogenesis": *Please, add references to the statements that start with "... During infection, T. cruzi moves through..." and end in "...The parasite exploits newly formed microvessels..."*

4) topic "Tackling Chagas Disease":

a) "Several approaches collectively provide a comprehensive view of the status and future directions in the fight against Chagas disease.": *if this statement does not refer directly to what is described in the subsequent paragraphs, please add a reference.*

b) "...including automated phenotypic screening and sensitive in vivo models (Carolina Borsoi Moraes, Universidade de São Paulo, São Paulo, Brazil).": *is there a reference to this work or any previous related research that has been published by this author?*

c) "Phenotypic drug discovery remains a cornerstone...": *please add a reference.*

d) "TESEO clinical trial...": *please provide the meaning of acronym TESEO.*

5) topic "Metabolic Adaptability of Trypanosoma cruzi"

a) "Marcia Cristina Paes (Universidade Estadual do Rio de Janeiro, Rio de Janeiro, Brazil)... ...Her study revealed that under glucose-rich conditions,": *Please provide a reference for this work, in case it has already been published.*

6) topic "Key Insights into Trypanosoma cruzi Biology"

a) "...Sergio Schenkman (Universidade Federal de São Paulo, São Paulo, Brazil) shared findings from single-cell RNA expression analysis findings...": *Please provide a reference for this work, in case it has already been published.*

b) "...Irina Afasizheva (University of California, Irvine, USA) explored mitochondrial RNA processing in trypanosomes...": *Please provide a reference for this work, in case it has already been published.*

c) "...José Roberto Sotelo-Silveira (Universidad de la República, Montevideo, Uruguay) went deep into the translational machinery of T. cruzi, showing that ribosomal protein (RP) mRNA translation is globally repressed...": *Please provide a reference for this work, in case it has already been published.*

d) "For instance, replicative epimastigotes differentiate into non-proliferative metacyclic trypomastigotes (metacyclogenesis) when nutrients are scarce, and growth is impaired...": *Please provide a reference for this statement.*

e) "Findings from Simone Guedes Calderano (Instituto Butantan, São Paulo, Brazil) showed that stationary epimastigotes are predominantly in the G1 phase,...": *Please provide a reference for this work, in case it has already been published.*

f) topic "Bridging Science and Society: Chagas Disease Awareness Initiatives"

a)"For the first time in its 50-year history,...": *please, mention that it is the Caxambu meeting history.*

b) "Integra Chagas Brasil Project": *is there a website (or any digital media) for this project? If yes, please add a reference to the link.*

c) "Brazilian Ministry of Health's Brazil Saudável Program ": *please please add a reference to this program link.*

d) "In an interactive game developed by Eduardo Caio Torres dos Santos (Instituto Oswaldo Cruz, Rio de Janeiro, Brazil)...": *is there a website (or any digital media) for this project? If yes, please add a reference to the link.*

e)"Museu de Saúde Pública do Emilio Ribas (MUSPER, São Paulo, Brazil)": *please add a reference to the website link.*

f) "the financial support from "Loccus do Brasil"": *please add a reference to the website link.*

7) topic "A Legacy of Scientific Excellence: The Research Contributions of Professor Erney Plessmann de Camargo":

a) "His research emphasized the importance of the One Health approach, integrating molecular epidemiology, phylogenetics, and taxonomy...": *Please, add some relevant references to Prof. Erney Camargo's research.*

b) "...in deciphering the genetic and biochemical foundations of this symbiosis...": *Please, add a reference here.*

8) topic "Historical Significance of the Chagas Disease Meeting":

a) "article by Goldenberg et al. (submitted)": *if the submitted article has been published please update the reference.*

b) "It is evident that research tools have significantly advanced over the past 50 years...": *I suggest removing this whole paragraph. It has to do more with the parasitology history in Brazil than to the Caxambu meeting and Chagas Disease, which is the subject matter of this Perspective.*

9) *I have not exhaustively searched for typos, but I've found some in the text. Please, revise the text and correct these ones:*

*a) Universiad Federal*

*b) Erney Cmargo*

Dear Dr. Adeilton Brandão,

We would like to thank you and the reviewers for the careful reading and thoughtful comments on our manuscript entitled " The 50th Anniversary Conference – Caxambu 2024" [MIOC-2025-0061], submitted to Memórias do Instituto Oswaldo Cruz.

We have revised the manuscript according to the suggestions provided. In particular, we have:

• Updated the title of the article to better reflect its scope and focus, as recommended in the review process.

• Included five of the ten most cited articles by Prof. Erney Plessmann Camargo, as indexed on Google Scholar, to better reflect the impact of his scientific contributions.

• Updated the reference to the article by Goldenberg et al., which has since been published and is now cited accordingly.

• Removed the paragraph suggested by Referee 2 (regarding the history of parasitology in Brazil) to maintain the focus on the Caxambu meeting and Chagas disease.

• Corrected all typographical errors and performed a full text revision to correct any remaining typos or inconsistencies.

• Added appropriate references where requested.

Please find enclosed our detailed responses to the reviewers' comments, along with the revised version of the manuscript.

We appreciate the opportunity to improve our work through the review process and look forward to your feedback.

Sincerely,

Renata Rosito Tonelli on behalf of all authors

This manuscript is a bit long for the "perpsective article" format currently adopted by Memórias do IOC. However, I have some suggestions to adjust the text as a "perspective article".

1) Topic "The 50th Anniversary Conference – Caxambu 2024":

a) I suggest removing the entire topic from the main text of article, and include the topic title into the main article title.

b) The paragraph "To celebrate 50 years of Chagas research, the Brazilian Society of Protozoology (SBPz) organized a four-day conference held in Caxambu, Minas Gerais, Brazil, from November 3 to 7, 2024. The meeting gathered world-leading experts from multiple disciplines whose research has expanded the boundaries of Chagas Disease studies." Please, add this paragraph to the abstract

Reply: Thank you for your suggestion. The topic "The 50th Anniversary Conference – Caxambu 2024" has been removed from the main text, and its subtitle has been incorporated as the main article title. Additionally, the paragraph has been included in the abstract as recommended.

2) Benznidazole and Nifurtimox: Please add references to the original works proposing the use of these drugs in Chagas Disease therapy.

Reply: References to the original works proposing the use of Benznidazole and Nifurtimox in Chagas Disease therapy (refs 9 and 10) have been included as requested.

3) Topic "Mechanisms of T. cruzi Pathogenesis": Please, add references to the statements that start with "... During infection, T. cruzi moves through..." and end in "...The parasite exploits newly formed microvessels..."

Reply References have been added (refs 4 and 5) to support the statements within the section "Mechanisms of T. cruzi Pathogenesis as recommended.

4) Topic "Tackling Chagas Disease":

a) "Several approaches collectively provide a comprehensive view of the status and future directions in the fight against Chagas disease.": if this statement does not refer directly to what is described in the subsequent paragraphs, please add a reference.

Reply: The reference "Chagas disease: current perspectives on a neglected tropical disease" has been included to support the statement regarding the comprehensive view of the status and future directions in the fight against Chagas disease.

b) "...including automated phenotypic screening and sensitive in vivo models (Carolina Borsoi Moraes, Universidade de São Paulo, São Paulo, Brazil).": is there a reference to this work or any previous related research that has been published by this author?

Reply: A reference (ref 14) to related research previously published by Carolina Borsoi Moraes on automated phenotypic screening and in vivo models has been included to address this comment.

c) "Phenotypic drug discovery remains a cornerstone...": please add a reference.

Reply: A reference supporting the statement "Phenotypic drug discovery remains a cornerstone in identifying potential treatments" has been included as reference [15].

d) "TESEO clinical trial...": please provide the meaning of acronym TESEO. OK

Reply: The meaning of the acronym TESEO has been provided in the text, as requested.

5) Topic "Metabolic Adaptability of Trypanosoma cruzi"

a) "Marcia Cristina Paes (Universidade Estadual do Rio de Janeiro, Rio de Janeiro, Brazil)... ...Her study revealed that under glucose-rich conditions,": Please provide a reference for this work, in case it has already been published.

Reply: A reference to the published work by Marcia Cristina Paes on glucose-rich conditions has been included [ref. 21], as requested.

6) Topic "Key Insights into Trypanosoma cruzi Biology"

a) "...Sergio Schenkman (Universidade Federal de São Paulo, São Paulo, Brazil) shared findings from single-cell RNA expression analysis findings...": Please provide a reference for this work, in case it has already been published.

Reply: The paragraph has been revised and a reference [22] has been added as requested:

b) "...Irina Afasizheva (University of California, Irvine, USA) explored mitochondrial RNA processing in trypanosomes...": Please provide a reference for this work, in case it has already been published.

Reply: Thank you for your suggestion. The paragraph has been revised to clarify that the findings are based on studies in Trypanosoma brucei. Additionally, the appropriate reference [23] has been included to support the information provided.

c) "...José Roberto Sotelo-Silveira (Universidad de la República, Montevideo, Uruguay) went deep into the translational machinery of T. cruzi, showing that ribosomal protein (RP) mRNA translation is globally repressed...": Please provide a reference for this work, in case it has already been published.

Reply: Thank you for your comment. A reference [24] has been added to support the statement regarding José Roberto Sotelo-Silveira's findings on the global repression of ribosomal protein mRNA translation in T. cruzi, as requested.

d) "For instance, replicative epimastigotes differentiate into non-proliferative metacyclic trypomastigotes (metacyclogenesis) when nutrients are scarce, and growth is impaired...": Please provide a reference for this statement.

Reply: Thank you for your observation. A reference [25] has been added to support the statement regarding metacyclogenesis induced by nutrient scarcity and growth impairment: Contreras, V. T., Salles, J. M., Thomas, N., Morel, C. M., & Goldenberg, S. (1985). In vitro differentiation of Trypanosoma cruzi under chemically defined conditions. Molecular and Biochemical Parasitology, 16(3), 315–327. https://doi.org/10.1016/0166-6851(85)90140-4

This seminal study demonstrates that T. cruzi epimastigotes undergo differentiation into metacyclic trypomastigotes under nutrient-deprived and stress conditions, as described.

e) "Findings from Simone Guedes Calderano (Instituto Butantan, São Paulo, Brazil) showed that stationary epimastigotes are predominantly in the G1 phase,...": Please provide a reference for this work, in case it has already been published.

Reply: Thank you for your comment. The study by Simone Guedes Calderano has been accepted and is currently available in its provisional form in Frontiers in Cellular and Infection Microbiology. Although the final formatted version is pending publication, we have included the citation with its DOI, as the content is publicly accessible and citable.

f) Topic "Bridging Science and Society: Chagas Disease Awareness Initiatives"

a)"For the first time in its 50-year history,...": please, mention that it is the Caxambu meeting history.

Reply: Thank you for your suggestion. The text has been revised to explicitly mention that the event refers to the history of the Caxambu meeting.

b) "Integra Chagas Brasil Project": is there a website (or any digital media) for this project? If yes, please add a reference to the link.

Reply: Thank you for your comment. The Integra Chagas Brasil Project does have an official digital presence. A reference to the project website has been added to the manuscript to provide readers with direct access to further information.

c) " Brazilian Ministry of Health's Brazil Saudável Program ": please please add a reference to this program link.

Reply: Thank you for your comment. A reference to the Brazil Saudável Program by the Brazilian Ministry of Health has been added to the manuscript to provide direct access to the official information. Link added: https://www.gov.br/saude/pt-br/composicao/saps/brasil-saudavel

d) "In an interactive game developed by Eduardo Caio Torres dos Santos (Instituto Oswaldo Cruz, Rio de Janeiro, Brazil)...": is there a website (or any digital media) for this project? If yes, please add a reference to the link.

Reply: Thank you for your suggestion. A digital link to the interactive game developed by Eduardo Caio Torres dos Santos has been added to the manuscript to provide access to the project. Link included: https://expedicaochagas.fiocruz.br

e)"Museu de Saúde Pública do Emilio Ribas (MUSPER, São Paulo, Brazil)": please add a reference to the website link.

Reply: Thank you for your suggestion. The official website link for the Museu de Saúde Pública do Emílio Ribas (MUSPER) has been added to the manuscript.

f) "the financial support from "Loccus do Brasil"": please add a reference to the website link.

Reply: Thank you for your comment. The website link for Loccus do Brasil, which provided financial support, has been added to the manuscript.

7) Topic "A Legacy of Scientific Excellence: The Research Contributions of Professor Erney Plessmann de Camargo":

a) "His research emphasized the importance of the One Health approach, integrating molecular epidemiology, phylogenetics, and taxonomy...": Please, add some relevant references to Prof. Erney Camargo's research.

Reply:. We appreciate the referee's suggestion. Accordingly, we have now included five articles [refs 30-34] that rank among the ten most cited publications by Prof. Erney Plessmann Camargo, as listed on Google Scholar.

b) "...in deciphering the genetic and biochemical foundations of this symbiosis...": Please, add a reference here.

Reply: As requested, we have included a relevant reference [42] addressing the genetic and biochemical foundations of this symbiosis:

8) topic "Historical Significance of the Chagas Disease Meeting":

a) "article by Goldenberg et al. (submitted)": if the submitted article has been published please update the reference.

Reply: We thank the referee for pointing this out. The previously cited article by Goldenberg et al., which was marked as "submitted," has now been updated to reflect its published version [ref 44].

b) "It is evident that research tools have significantly advanced over the past 50 years...": I suggest removing this whole paragraph. It has to do with the parasitology history in Brazil than to the Caxambu meeting and Chagas Disease, which is the subject matter of this Perspective.

Reply: We agree with the referee's observation. The paragraph in question has been removed from the revised version of the manuscript.

9) I have not exhaustively searched for typos, but I've found some in the text. Please, revise the text and correct these ones:

a) Universiad Federal

b) Erney Cmargo

Reply: We thank the referee for pointing out the typographical errors. The suggested corrections have been made — "Universiad Federal" and "Erney Cmargo" have been corrected to "Universidade Federal" and "Erney Camargo," respectively. A thorough revision of the text was also performed to identify and fix additional typos.

## SECOND REVIEW ROUND

### REVIEWERS' COMMENTS

**REVIEWER #1**

No comments.

