## [Reviewer Report · FIRST REVIEW ROUND - REVIEWERS COMMENTS]

## REVIEWER #1

This manuscript is a bit long for the “perpsective article” format currently adopted by Memórias do IOC. However, I have some suggestions to adjust the text as a “perspective article”.

1) topic “The 50th Anniversary Conference - Caxambu 2024”:

a) I suggest removing the entire topic from the main text of article, and include the topic title into the main article title.

b) The paragraph “To celebrate 50 years of Chagas research, the Brazilian Society of Protozoology (SBPz) organized a four-day conference held in Caxambu, Minas Gerais, Brazil, from November 3 to 7, 2024. The meeting gathered world-leading experts from multiple disciplines whose research has expanded the boundaries of Chagas Disease studies.” Please, add this paragraph to the abstract.

2) Benznidazole and Nifurtimox: Please add references to the original works proposing the use of these drugs in Chagas Disease therapy.

3) topic “Mechanisms of T. cruzi Pathogenesis”: Please, add references to the statements that start with “... During infection, T. cruzi moves through...” and end in “...The parasite exploits newly formed microvessels...”

4) topic “Tackling Chagas Disease”:

a) “Several approaches collectively provide a comprehensive view of the status and future directions in the fight against Chagas disease.”: if this statement does not refer directly to what is described in the subsequent paragraphs, please add a reference.

b) “...including automated phenotypic screening and sensitive in vivo models (Carolina Borsoi Moraes, Universidade de São Paulo, São Paulo, Brazil).”: is there a reference to this work or any previous related research that has been published by this author?

c) “Phenotypic drug discovery remains a cornerstone...”: please add a reference.

d) “TESEO clinical trial...”: please provide the meaning of acronym TESEO.

5) topic “Metabolic Adaptability of Trypanosoma cruzi”

a) “Marcia Cristina Paes (Universidade Estadual do Rio de Janeiro, Rio de Janeiro, Brazil)... ...Her study revealed that under glucose-rich conditions,”: Please provide a reference for this work, in case it has already been published.

6) topic “Key Insights into Trypanosoma cruzi Biology”

a) “...Sergio Schenkman (Universidade Federal de São Paulo, São Paulo, Brazil) shared findings from single-cell RNA expression analysis findings...”: Please provide a reference for this work, in case it has already been published.

b) “...Irina Afasizheva (University of California, Irvine, USA) explored mitochondrial RNA processing in trypanosomes...”: Please provide a reference for this work, in case it has already been published.

c) “...José Roberto Sotelo-Silveira (Universidad de la República, Montevideo, Uruguay) went deep into the translational machinery of T. cruzi, showing that ribosomal protein (RP) mRNA translation is globally repressed...”: Please provide a reference for this work, in case it has already been published.

d) “For instance, replicative epimastigotes differentiate into non-proliferative metacyclic trypomastigotes (metacyclogenesis) when nutrients are scarce, and growth is impaired...”: Please provide a reference for this statement.

e) “Findings from Simone Guedes Calderano (Instituto Butantan, São Paulo, Brazil) showed that stationary epimastigotes are predominantly in the G1 phase,...”: Please provide a reference for this work, in case it has already been published.

f) topic “Bridging Science and Society: Chagas Disease Awareness Initiatives”

a)”For the first time in its 50-year history,...”: please, mention that it is the Caxambu meeting history.

b) “Integra Chagas Brasil Project”: is there a website (or any digital media) for this project? If yes, please add a reference to the link.

c) “Brazilian Ministry of Health’s Brazil Saudável Program “: please please add a reference to this program link.

d) “In an interactive game developed by Eduardo Caio Torres dos Santos (Instituto Oswaldo Cruz, Rio de Janeiro, Brazil)...”: is there a website (or any digital media) for this project? If yes, please add a reference to the link.

e)”Museu de Saúde Pública do Emilio Ribas (MUSPER, São Paulo, Brazil)”: please add a reference to the website link.

f) “the financial support from “Loccus do Brasil””: please add a reference to the website link.

7) topic “A Legacy of Scientific Excellence: The Research Contributions of Professor Erney Plessmann de Camargo”:

a) “His research emphasized the importance of the One Health approach, integrating molecular epidemiology, phylogenetics, and taxonomy...”: Please, add some relevant references to Prof. Erney Camargo’s research.

b) “...in deciphering the genetic and biochemical foundations of this symbiosis...”: Please, add a reference here.

8) topic “Historical Significance of the Chagas Disease Meeting”:

a) “article by Goldenberg et al. (submitted)”: if the submitted article has been published please update the reference.

b) “It is evident that research tools have significantly advanced over the past 50 years...”: I suggest removing this whole paragraph. It has to do more with the parasitology history in Brazil than to the Caxambu meeting and Chagas Disease, which is the subject matter of this Perspective.

9) I have not exhaustively searched for typos, but I’ve found some in the text. Please, revise the text and correct these ones:

a) Universiad Federal

b) Erney Cmargo

---

## [Author Response · AUTHORS RESPONSE TO REVIEWERS]

## AUTHORS’ RESPONSE TO THE REVIEWERS

Dear Dr. Adeilton Brandão,

We would like to thank you and the reviewers for the careful reading and thoughtful comments on our manuscript entitled “ The 50th Anniversary Conference – Caxambu 2024” [MIOC-2025-0061], submitted to Memórias do Instituto Oswaldo Cruz.

We have revised the manuscript according to the suggestions provided. In particular, we have:

• Updated the title of the article to better reflect its scope and focus, as recommended in the review process.

• Included five of the ten most cited articles by Prof. Erney Plessmann Camargo, as indexed on Google Scholar, to better reflect the impact of his scientific contributions.

• Updated the reference to the article by Goldenberg et al., which has since been published and is now cited accordingly.

• Removed the paragraph suggested by Referee 2 (regarding the history of parasitology in Brazil) to maintain the focus on the Caxambu meeting and Chagas disease.

• Corrected all typographical errors and performed a full text revision to correct any remaining typos or inconsistencies.

• Added appropriate references where requested.

Please find enclosed our detailed responses to the reviewers’ comments, along with the revised version of the manuscript.

We appreciate the opportunity to improve our work through the review process and look forward to your feedback.

Sincerely,

Renata Rosito Tonelli on behalf of all authors

---

## [Author Response · ANSWERS TO REVIEWERS]

## ANSWERS TO REVIEWERS

This manuscript is a bit long for the “perpsective article” format currently adopted by Memórias do IOC. However, I have some suggestions to adjust the text as a “perspective article”.

1) Topic “The 50th Anniversary Conference – Caxambu 2024”:

a) I suggest removing the entire topic from the main text of article, and include the topic title into the main article title.

b) The paragraph “To celebrate 50 years of Chagas research, the Brazilian Society of Protozoology (SBPz) organized a four-day conference held in Caxambu, Minas Gerais, Brazil, from November 3 to 7, 2024. The meeting gathered world-leading experts from multiple disciplines whose research has expanded the boundaries of Chagas Disease studies.” Please, add this paragraph to the abstract

Reply: Thank you for your suggestion. The topic “The 50th Anniversary Conference – Caxambu 2024” has been removed from the main text, and its subtitle has been incorporated as the main article title. Additionally, the paragraph has been included in the abstract as recommended.

2) Benznidazole and Nifurtimox: Please add references to the original works proposing the use of these drugs in Chagas Disease therapy.

Reply: References to the original works proposing the use of Benznidazole and Nifurtimox in Chagas Disease therapy (refs 9 and 10) have been included as requested.

3) Topic “Mechanisms of T. cruzi Pathogenesis”: Please, add references to the statements that start with “... During infection, T. cruzi moves through...” and end in “...The parasite exploits newly formed microvessels...”

Reply References have been added (refs 4 and 5) to support the statements within the section “Mechanisms of T. cruzi Pathogenesis as recommended.

4) Topic “Tackling Chagas Disease”:

a) “Several approaches collectively provide a comprehensive view of the status and future directions in the fight against Chagas disease.”: if this statement does not refer directly to what is described in the subsequent paragraphs, please add a reference.

Reply: The reference “Chagas disease: current perspectives on a neglected tropical disease” has been included to support the statement regarding the comprehensive view of the status and future directions in the fight against Chagas disease.

b) “...including automated phenotypic screening and sensitive in vivo models (Carolina Borsoi Moraes, Universidade de São Paulo, São Paulo, Brazil).”: is there a reference to this work or any previous related research that has been published by this author?

Reply: A reference (ref 14) to related research previously published by Carolina Borsoi Moraes on automated phenotypic screening and in vivo models has been included to address this comment.

c) “Phenotypic drug discovery remains a cornerstone...”: please add a reference.

Reply: A reference supporting the statement “Phenotypic drug discovery remains a cornerstone in identifying potential treatments” has been included as reference [15].

d) “TESEO clinical trial...”: please provide the meaning of acronym TESEO. OK

Reply: The meaning of the acronym TESEO has been provided in the text, as requested.

5) Topic “Metabolic Adaptability of Trypanosoma cruzi”

a) “Marcia Cristina Paes (Universidade Estadual do Rio de Janeiro, Rio de Janeiro, Brazil)... ...Her study revealed that under glucose-rich conditions,”: Please provide a reference for this work, in case it has already been published.

Reply: A reference to the published work by Marcia Cristina Paes on glucose-rich conditions has been included [ref. 21], as requested.

6) Topic “Key Insights into Trypanosoma cruzi Biology”

a) “...Sergio Schenkman (Universidade Federal de São Paulo, São Paulo, Brazil) shared findings from single-cell RNA expression analysis findings...”: Please provide a reference for this work, in case it has already been published.

Reply: The paragraph has been revised and a reference [22] has been added as requested:

b) “...Irina Afasizheva (University of California, Irvine, USA) explored mitochondrial RNA processing in trypanosomes...”: Please provide a reference for this work, in case it has already been published.

Reply: Thank you for your suggestion. The paragraph has been revised to clarify that the findings are based on studies in Trypanosoma brucei. Additionally, the appropriate reference [23] has been included to support the information provided.

c) “...José Roberto Sotelo-Silveira (Universidad de la República, Montevideo, Uruguay) went deep into the translational machinery of T. cruzi, showing that ribosomal protein (RP) mRNA translation is globally repressed...”: Please provide a reference for this work, in case it has already been published.

Reply: Thank you for your comment. A reference [24] has been added to support the statement regarding José Roberto Sotelo-Silveira’s findings on the global repression of ribosomal protein mRNA translation in T. cruzi, as requested.

d) “For instance, replicative epimastigotes differentiate into non-proliferative metacyclic trypomastigotes (metacyclogenesis) when nutrients are scarce, and growth is impaired...”: Please provide a reference for this statement.

Reply: Thank you for your observation. A reference [25] has been added to support the statement regarding metacyclogenesis induced by nutrient scarcity and growth impairment: Contreras, V. T., Salles, J. M., Thomas, N., Morel, C. M., Goldenberg, S. (1985). In vitro differentiation of Trypanosoma cruzi under chemically defined conditions. Molecular and Biochemical Parasitology, 16(3), 315–327. https://doi.org/10.1016/0166-6851(85)90140-4 This seminal study demonstrates that T. cruzi epimastigotes undergo differentiation into metacyclic trypomastigotes under nutrient-deprived and stress conditions, as described.

e) “Findings from Simone Guedes Calderano (Instituto Butantan, São Paulo, Brazil) showed that stationary epimastigotes are predominantly in the G1 phase,...”: Please provide a reference for this work, in case it has already been published.

Reply: Thank you for your comment. The study by Simone Guedes Calderano has been accepted and is currently available in its provisional form in Frontiers in Cellular and Infection Microbiology. Although the final formatted version is pending publication, we have included the citation with its DOI, as the content is publicly accessible and citable.

f) Topic “Bridging Science and Society: Chagas Disease Awareness Initiatives”

a)”For the first time in its 50-year history,...”: please, mention that it is the Caxambu meeting history.

Reply: Thank you for your suggestion. The text has been revised to explicitly mention that the event refers to the history of the Caxambu meeting.

b) “Integra Chagas Brasil Project”: is there a website (or any digital media) for this project? If yes, please add a reference to the link.

Reply: Thank you for your comment. The Integra Chagas Brasil Project does have an official digital presence. A reference to the project website has been added to the manuscript to provide readers with direct access to further information.

c) “ Brazilian Ministry of Health’s Brazil Saudável Program “: please please add a reference to this program link.

Reply: Thank you for your comment. A reference to the Brazil Saudável Program by the Brazilian Ministry of Health has been added to the manuscript to provide direct access to the official information. Link added: https://www.gov.br/saude/pt-br/composicao/saps/brasil-saudavel

d) “In an interactive game developed by Eduardo Caio Torres dos Santos (Instituto Oswaldo Cruz, Rio de Janeiro, Brazil)...”: is there a website (or any digital media) for this project? If yes, please add a reference to the link.

Reply: Thank you for your suggestion. A digital link to the interactive game developed by Eduardo Caio Torres dos Santos has been added to the manuscript to provide access to the project. Link included: https://expedicaochagas.fiocruz.br

e)”Museu de Saúde Pública do Emilio Ribas (MUSPER, São Paulo, Brazil)”: please add a reference to the website link.

Reply: Thank you for your suggestion. The official website link for the Museu de Saúde Pública do Emílio Ribas (MUSPER) has been added to the manuscript.

f) “the financial support from “Loccus do Brasil””: please add a reference to the website link.

Reply: Thank you for your comment. The website link for Loccus do Brasil, which provided financial support, has been added to the manuscript.

7) Topic “A Legacy of Scientific Excellence: The Research Contributions of Professor Erney Plessmann de Camargo”:

a) “His research emphasized the importance of the One Health approach, integrating molecular epidemiology, phylogenetics, and taxonomy...”: Please, add some relevant references to Prof. Erney Camargo’s research.

Reply:. We appreciate the referee’s suggestion. Accordingly, we have now included five articles [refs 30-34] that rank among the ten most cited publications by Prof. Erney Plessmann Camargo, as listed on Google Scholar.

b) “...in deciphering the genetic and biochemical foundations of this symbiosis...”: Please, add a reference here.

Reply: As requested, we have included a relevant reference [42] addressing the genetic and biochemical foundations of this symbiosis:

8) topic “Historical Significance of the Chagas Disease Meeting”:

a) “article by Goldenberg et al. (submitted)”: if the submitted article has been published please update the reference.

Reply: We thank the referee for pointing this out. The previously cited article by Goldenberg et al., which was marked as “submitted,” has now been updated to reflect its published version [ref 44].

b) “It is evident that research tools have significantly advanced over the past 50 years...”: I suggest removing this whole paragraph. It has to do more with the parasitology history in Brazil than to the Caxambu meeting and Chagas Disease, which is the subject matter of this Perspective.

Reply: We agree with the referee’s observation. The paragraph in question has been removed from the revised version of the manuscript.

9) I have not exhaustively searched for typos, but I’ve found some in the text. Please, revise the text and correct these ones:

a) Universiad Federal

b) Erney Cmargo

Reply: We thank the referee for pointing out the typographical errors. The suggested corrections have been made — “Universiad Federal” and “Erney Cmargo” have been corrected to “Universidade Federal” and “Erney Camargo,” respectively. A thorough revision of the text was also performed to identify and fix additional typos.